crystallography/materials science

metal-organic frameworks, MIL-121, crystal habit, hydrothermal synthesis

**Author for correspondence:**
Liang Zhu
e-mail: zhuliang@tust.edu.cn

This article has been edited by the Royal Society of Chemistry, including the commissioning, peer review process and editorial aspects up to the point of acceptance.

# Research on the effects of hydrothermal synthesis conditions on the crystal habit of MIL-121

Fei Wang, Liang Zhu, Qingyan Wei and Yanfei Wang

College of Chemical Engineering and Materials, Tianjin University of Science and Technology, Tianjin 300457, People's Republic of China

FW, 0000-0001-7046-8627; LZ, 0000-0002-9086-4202

The investigation of the influence of hydrothermal synthesis conditions such as synthetic temperature, the amount of solvent, addition of additives including sodium hydroxide, lithium chloride and 2-methylimidazole on the morphology and size of MIL-121 was carried out. The experimental results indicated the significant impact of hydrothermal synthesis conditions on the morphologies and sizes of MIL-121 crystals. The synthesis temperature has little effect on the morphology, which is mainly reflected in the change of the aspect ratio, but the effect on the size is significant under low-temperature conditions. Additives have an important influence on the morphology and size of MIL-121. Our study provides a potential for the improvement of MIL-121 adsorption performance.

## 1. Introduction

Metal-organic frameworks (MOFs) have attracted wide attention in recent years because of their adjustable structure and porous size which attribute many valuable properties to the MOFs materials. Some studies have shown that the crystal habit of MOFs has an important relationship with its application performance in various research fields [1]. The morphology and size of MOFs are crucial to its performance improvement. Although there are many kinds of MOFs with controllable composition, there is less concern about the control of MOFs crystal habit. The design of crystal morphology is an indispensable part of expanding the application of crystalline materials [2]. So far, there are many methods for the regulation of the morphology and size of MOFs. For instance, Ghorbanloo and his co-workers used pyridine and acetic acid as modulators to prepare TMU-22 microbars and micro-plates via the hydrothermal method. The influence of reaction temperature, solvent and regulator concentration on the

synthesis of micro-crystals with different morphologies and sizes were elucidated, respectively [3]. Duan et al. [4] reported the synthesis of three hierarchical porous MOFs with adjustable morphology and porosity using organic amines as the template agent in the research system. These hierarchical porous MOFs have a hierarchy of micropore, mesopore and macropore and high thermal stability after simple washing and drying. However, the traditional method has a serious disadvantage, in that the modulator and template agent are difficult to remove, and it is urgent to develop a fast and green regulation strategy.

The reaction temperature, cooling rate and mixing solvent ratio of the adjustment system are the preferred methods to control the morphology and size. For example, spherical MIL-101 crystal was prepared by the hydrothermal method under the condition that the reaction system temperature was adjusted to 150°C [5]. $NH_2$-MIL-53 crystals with different morphologies were obtained by changing the proportion of water in the mixed solvent [6]. With the application of microwave, ultrasonic and other technologies in the field of materials, more rapid and environmentally friendly methods of synthesizing MOFs are beginning to emerge. The reported methods include the microwave auxiliary method [7–9], the sound wave induction method [10,11], the laser induction method [12], the gravity method [13], the spray drying method [14] and the ionic liquid microemulsion method [15]. These methods have been used to prepare nano-MOFs with controlled shape and size to achieve improvements in performance. However, there are few studies on the preparation of large particle size MOF.

In the industrial application process, the different morphology of MOFs has a significant impact on its stability and fluidity, which is widely used in core-shell MOFs synthesis [16], gas selective separation [17,18], nanomaterials [19,20], electrochemistry [21], photocatalysis [22], electro-catalysis [23], drug delivery [24,25] and other fields. Similarly, the size of the material is closely related to its performance and application. Zhang et al. optimized catalytic performance by controlling the size of the MOF-immobilized metal nanoparticles [26]. Consequently, the preparation of MOFs with different shapes and sizes has received extensive attention. MIL-121 is a typical type of thermally responsive amphoteric MOFs synthesized by hard acid ($Al^{3+}$) and pyromellitic acid ligands, which was first reported by Loiseau et al. [27] using the hydrothermal method at 210°C. It has been applied to metal salt adsorption [28], $Ln^{3+}$-doped fluorescence MOF preparation [29], $Ag^+$ selective detection [30], separation of hydrogen [31] and other fields. However, there were no reports on the morphology and size control of MIL-121. Furthermore, there were few studies focusing on the mechanism and the formation of a crystal habit of MOFs under hydrothermal reaction conditions.

According to the literature [32], the surface area of MIL-121 is only 162 $m^2 g^{-1}$, which is extremely detrimental to its application to gas adsorption separation. Chen et al. [33] suggested that the thermal trigger decarboxylation method obtained hierarchical porous MOF to improve the adsorption capacity of MIL-121 to $CO_2$, $C_2H_2$, $C_2H_4$ and $CH_4$ gases. In this paper, the control of MIL-121 morphology and size was investigated by adjusting the hydrothermal reaction conditions. The crystallization mechanism of MIL-121 was analysed from the crystallography, which provides the possibility for the enhancement of MIL-121 metal salt adsorption capacity. At the same time, it also provides a research strategy for the morphology, size control method and mechanism of other types of MOFs.

# 2. Material and methods

## 2.1. Materials and chemicals

Pyromellitic acid ($H_4$BTEC) was purchased from Sain Chemical Technology Inc. (Shanghai, China). Aluminium nitrate nonahydrate ($Al(NO_3)_3 \cdot 9H_2O$) was supplied by Tianjin Ailian Electronic Technology Inc. Sodium hydroxide was purchased from Sinopharm Group Chemical Reagent Co., Ltd. The chemical reagent including lithium chloride and ethanol were obtained from Shanghai Meirrell Chemical Technology Co., Ltd. 2-Methylimidazole was supplied by Icano Technology Co., Ltd. (Beijing, China). Deionized water was purchased from the Salt Science and Membrane Technology Research Group of Tianjin University of Science and Technology.

## 2.2. Synthesis at different reaction temperatures

MIL-121(Al) crystals were synthesized by the hydrothermal method. In this system, $Al(NO_3)_3 \cdot 9H_2O$ was used as the metal source, and pyromellitic acid was used as the organic linker. In the synthesis step, $Al(NO_3)_3 \cdot 9H_2O$ (2.40 g, 3.2 mmol) and $H_4$BTEC (0.80 g, 1.6 mmol) were both dissolved in a beaker

containing 10 ml of deionized water. After stirring for 10 min under the condition of magnetic stirring with temperature controlled at 60°C, a clear solution was attained. Five identical mixed solutions were prepared and transferred in five polyphenyl-lined autoclaves (100 ml). After that, they were reacted at 120, 150, 180, 210 and 240°C, respectively for 48 h. After the reaction was completed, the crystals in the five reactors were washed by the same centrifugal method. That was washed three times with ethanol and then several times with deionized water, followed by drying at 80°C before further characterization.

## 2.3. Synthesis at different amounts of solvent

$Al(NO_3)_3 \cdot 9H_2O$ (2.40 g, 3.2 mmol) and $H_4BTEC$ (0.80 g, 1.6 mmol) were dissolved in five beakers containing 10, 20, 30, 40 and 50 ml deionized water. The solution was then maintained under magnetic stirring at 60°C for several hours. After that, the solution was transferred into the polyphenyl-lined reactor with a capacity of 50 ml and then reacted in a programmed temperature-controlled oven at 150°C for 48 h. Subsequently, the autoclave was lowered to room temperature at a slow cooling rate of 5°C h$^{-1}$. The white crystals were obtained by centrifugation, subsequently washed with deionized water five times, followed by drying at 80°C before further characterization.

## 2.4. Synthesis at different amounts of sodium hydroxide

$Al(NO_3)_3 \cdot 9H_2O$ (2.40 g, 3.2 mmol) and $H_4BTEC$ (0.80 g, 1.6 mmol) were dissolved in a beaker containing 10 ml of deionized water under stirring to prepare mixed solution A, and accurately weighed 16 g NaOH to prepare 4 mol l$^{-1}$ NaOH standard solution B. Then 2, 4, 8 and 16 mmol solution B (4 mol l$^{-1}$ NaOH solution) were slowly added to the solution A to form mixed solution C, the mixed solution C was maintained with magnetic stirring for several hours at 70°C. Subsequently, the mixed solution C was transferred to 50 ml of polyphenyl-lined reactor and reacted in a programmed temperature-controlled oven at 150°C for 48 h. After that, the autoclave was cooled to 25°C at a programmed cooling rate of 5°C h$^{-1}$. The pH of the system before and after the reaction was measured. The white crystals were obtained by centrifugation, subsequently washed with deionized water five times, followed by drying at 80°C before further characterization.

## 2.5. Synthesis at different amounts of lithium chloride

$Al(NO_3)_3 \cdot 9H_2O$ (2.40 g, 3.2 mmol) and $H_4BTEC$ (0.80 g, 1.6 mmol) were dissolved in a beaker containing 10 ml of deionized water under stirring to prepare mixed solution A and accurately weighed 16.8 g LiCl to prepare 4 mol l$^{-1}$ LiCl standard solution B. Then 4, 8, 16 and 32 mmol solution B (4 mol l$^{-1}$ LiCl solution) were slowly added to the solution A, respectively; to prepare solution C, the mixed solution C was maintained for several hours at 70°C with magnetic stirring. Then, solution C was transferred into the polyphenyl-lined reactor with the specific capacity of 50 ml, which was reacted in a programmed temperature-controlled oven at 150°C for 48 h. After that, the autoclave was cooled to 25°C at a programmed cooling rate of 5°C h$^{-1}$. The white crystals were obtained by centrifugation, subsequently washed with deionized water five times, followed by drying at 80°C before further characterization.

## 2.6. Synthesis at different amounts of 2-methylimidazole

$Al(NO_3)_3 \cdot 9H_2O$ (2.40 g, 3.2 mmol), $H_4BTEC$ (0.80 g, 1.6 mmol) and deionized water (10 ml) were mixed in a polyphenyl-lined autoclave (100 ml), then 4, and 8 mmol 2-methylimidazole (2-MI) were added, respectively. After that, the autoclave was heated in a programmed temperature controlled oven at 150°C for 48 h. After that, the reaction kettle was cooled to 25°C at a programmed cooling rate of 5°C h$^{-1}$. The products were obtained by centrifugation, subsequently washed with deionized water five times, followed by drying at 80°C before further characterization.

## 2.7. Characterization methods

In order to confirm the crystal form and polymorphic purity of the products, powder X-ray diffraction (PXRD) patterns were recorded on a Shimadzu PXRD-6100 diffractometer equipped with Cu K$\alpha$ source at a scanning rate of 5° min$^{-1}$. The crystal habit of MIL-121 was determined by a JSM-6380LV scanning electron microscope (SEM). The size of the crystals was characterized using a laser

diffraction particle size analyser (Beckman Coulter LS) at a pump speed of 52 and ultrasonic time 5 s. The Fourier transform-infrared (FT-IR) spectra (potassium bromide pellets) were obtained on a TENSOR27 (Bruker, Germany) FT-IR spectrometer instrument in the wavenumber range of 400–4000 cm$^{-1}$. The Raman detection was carried out on a LabRAM HR800 microprobe Raman system (Horiba Jobin-Yvon) with excitation of 532 nm. The Raman band of a silicon wafer at 520 cm$^{-1}$ was used as a reference to calibrate the spectrometer. The Raman shift range is 250–4000 cm$^{-1}$ and the exposure time is 5 s. The thermogravimetric data were collected by a SDT-Q600 differential of American TA Company performed in the temperature range of 40–800°C, with a temperature scanning rate of 20°C min$^{-1}$ in 50 ml min$^{-1}$ N$_2$ gas flow.

# 3. Results and discussion

## 3.1. Effect of the synthesis temperature

MIL-121 crystals were prepared at different reaction temperatures by the hydrothermal method. Figure 1 depicts the SEM images of crystals prepared under different synthesis temperature conditions. The result shows that temperature has no significant effect on crystal habits but the size of the crystals has a notable change from figure 1a–c. At the same time, the aspect ratio of crystals decreases from figure 1b,c. Because the four reaction temperatures use the same heating rate, reaction time, cooling rate and solvent ratio, it can be seen that the reaction temperature mainly influences the particle size of MIL-121 crystals, but has little effect on crystal habit. As shown in figure 3, the crystals obtained above 180°C pose the different crystal form (MIL-118), thus figure 1d cannot be compared with others owing to the change in crystal form. Figure 2 demonstrates the size distribution of products under different operating conditions. The results show that the particle size distribution below 180°C obeys the typical normal distribution and the particle size reaches the maximum at 180°C with the increase of temperature. At 210°C, the crystal size distribution appears in a dual model distribution, because when the temperature was higher than 180°C, the crystal form of MIL-121 was transformed into MIL-118, and the product existed in the form of a mixture of the two. This provides a temperature control range for the subsequent MIL-121 crystal habit control in this article.

Figure 3a is the PXRD pattern of crystal products obtained under the synthesis performed at different temperatures. The MIL-121 pure phases appeared at low temperatures of 180°C and below by comparison with the PXRD pattern provided by the crystallographic information file [32] in the Cambridge Crystallographic Data Centre database. When the reaction temperature approached 210°C, it converted to the mixing phase of MIL-121 and MIL-118 (MIL-121 is primary phase). The peak of MIL-121 at around 8.48° is relatively weaker compared to the peak of MIL-118. The peaks of MIL-118 appeared at around 10.04°, 15.0°, 15.8° and 18.36°. The main peaks of the MIL-121 phase disappears completely and converts to MIL-118 phase at 240°C, which can be compared with the PXRD pattern provided by the crystallographic information file [34]. Figure 3b shows the FT-IR of crystal products obtained under different reaction temperature conditions. It can be seen that the infrared spectra of products under the conditions of 120°C, 150°C and 180°C are completely consistent with the reported of MIL-121 [32]. However, the infrared spectra of the particles obtained under the conditions of 210°C, 240°C have a significant difference, in which the peaks of the free carboxyl functional group at 2524.7 cm$^{-1}$, 2663.6 cm$^{-1}$ disappeared.

## 3.2. Effects of solvent, sodium hydroxide, lithium chloride and 2-methylimidazole

Five kinds of MIL-121 crystals with different crystal habit and size were prepared by regulating the hydrothermal reaction conditions. Figures 4 and 5 are SEM pictures of the crystals synthesized with different hydrothermal reaction conditions, from which we can see that there are five different morphologies; including irregular cubic, irregular block, octahedron, column and stacked sheet. First of all, when using a solvent with the amount of 40 ml, the product was an irregular cubic crystal and the particle sizes are uneven as shown in figure 4d. It is the preferred result of the synthesis of MIL-121 under the conditions 10, 20, 30, 40 and 50 ml solvents, respectively. The experimental results show that as the solvent ratio increases, the morphologies of MIL-121 gradually transformed from a parallelepiped to an irregular cubic crystal, with particle size distributed from 2 to 40 µm. The product has a larger particle size and a smaller aspect ratio. With the 4 mol l$^{-1}$ sodium hydroxide solution with the amount of 8 mmol, the product morphology grows into irregularly blocky shapes and has a large crystal size as shown in figure 4h. It was the best result of the synthesis of MIL-121 morphology under the conditions

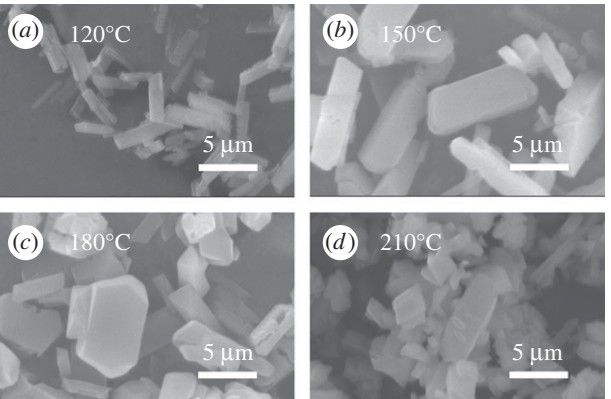

**Figure 1.** The SEM images of MIL-121 prepared under different synthesis temperatures: (*a*) 120°C, (*b*) 150°C, (*c*) 180°C, and (*d*) 210°C.

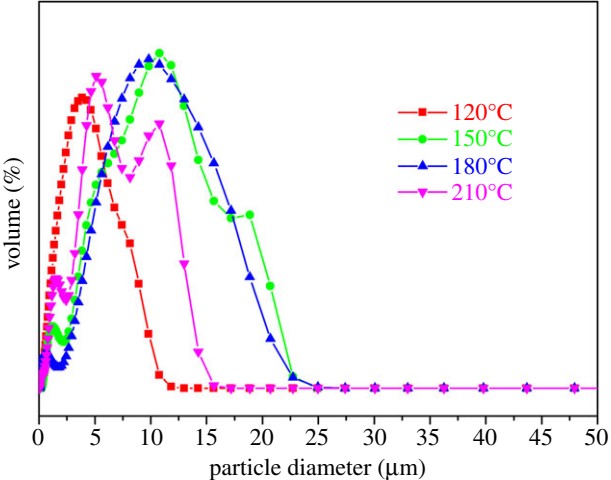

**Figure 2.** The size distribution images of MIL-121 prepared at different temperatures.

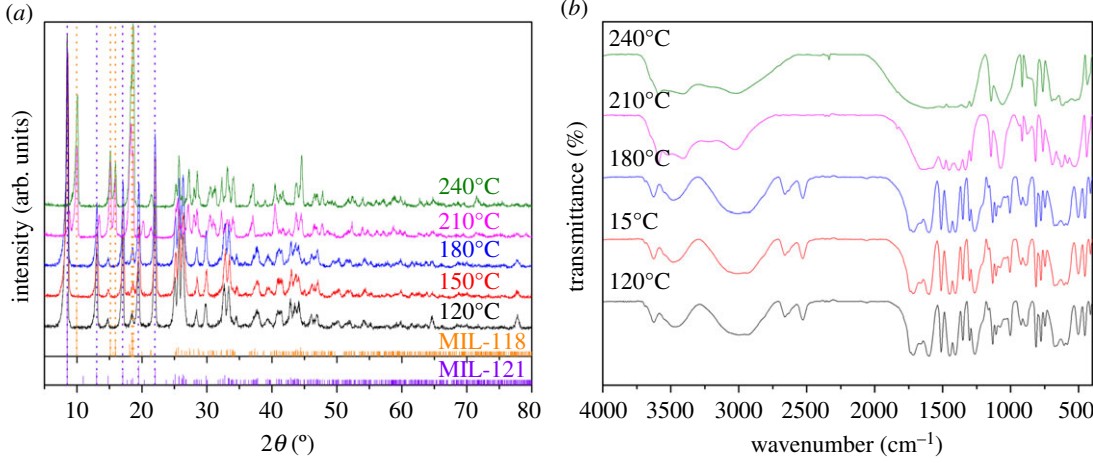

**Figure 3.** The PXRD and FT-IR images of MIL-121 prepared at different temperatures: (*a*) PXRD, (*b*) FT-IR.

of 4, 8 and 16 mmol, respectively. The characterization results show that the optimal addition amount of sodium hydroxide is 8 mmol and the crystal products were irregular block shapes. The pH value of the system before the reaction was 1.67, and that after the reaction was 1.72. When the addition amount of sodium hydroxide was more than 8 mmol, the crystal was agglomerated after centrifugation and drying.

The product prepared by the addition of lithium chloride with the amount of 32 mmol was column shaped with uneven size as shown in figure 5*d*, which was the preferred morphology of MIL-121

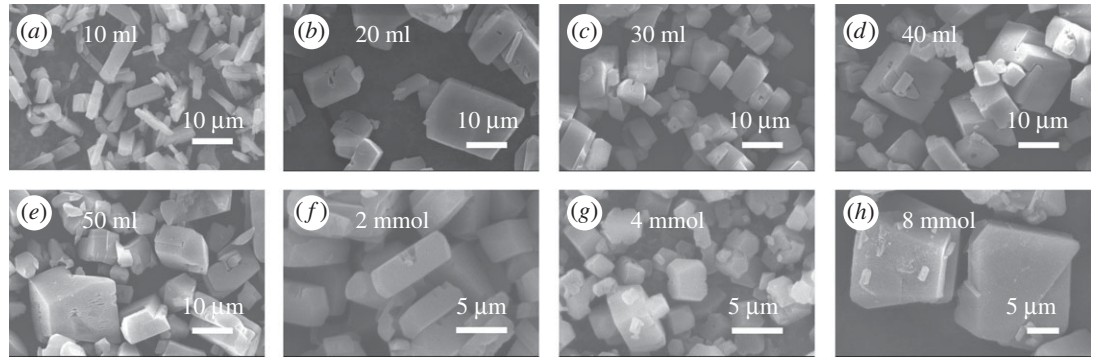

**Figure 4.** The SEM images of MIL-121 prepared at different conditions: (*a*) solvent 10 ml, (*b*) solvent 20 ml, (*c*) solvent 30 ml, (*d*) solvent 40 ml, (*e*) solvent 50 ml, (*f*) NaOH 2 mmol, (*g*) NaOH 4 mmol, (*h*) NaOH 8 mmol.

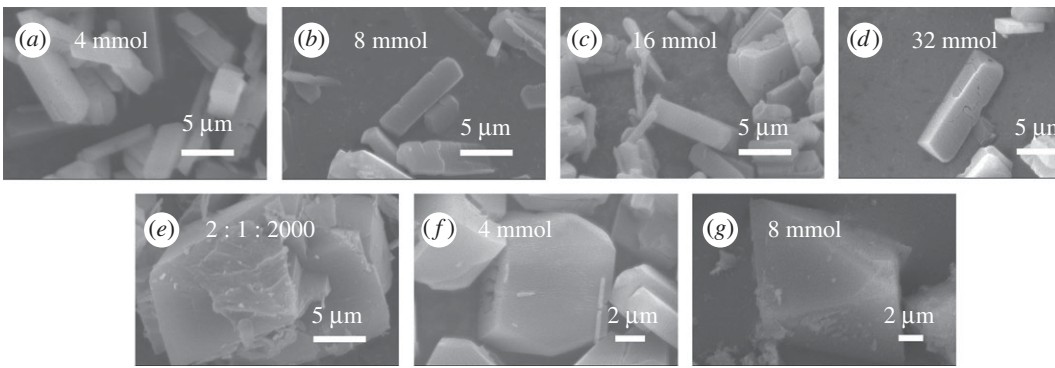

**Figure 5.** The SEM images of MIL-121 prepared at different conditions: (*a*) LiCl 4 mmol, (*b*) LiCl 8 mmol, (*c*) LiCl 16 mmol, (*d*) LiCl 32 mmol, (*e*) Al(NO3)$_3 \cdot$ 9H$_2$O:H$_4$BTEC:H$_2$O = 2 : 1 : 2000, (*f*) 2-MI 4 mmol and (*g*) 2-MI 8 mmol.

synthetic with the addition of 4, 8, 16 and 32 mmol aqueous solutions of lithium chloride, respectively. The characterization results indicated that lithium chloride will cause the crystal aspect ratio to become larger, and the product shape change from a parallelepiped to a column. The product prepared by the addition of 2-MI with the amount of 8 mmol was a octahedral shaped with uneven size figure 5*g*, which was the preferred result of MIL-121 synthetic morphology under different 2-MI addition conditions. The characterization results showed that the addition of 2-MI leads to a smaller aspect ratio and larger particles. Figure 5*e* is the SEM image of a crystal obtained under the condition of raw ratio Al(NO$_3$)$_3 \cdot$ 9H$_2$O : H$_4$BTEC : H$_2$O = 2 : 1 : 2000, which was a stacked sheet and uneven in size.

Figure 6*a* is the PXRD pattern of the MIL-121 synthesized under different solvent ratios. When the solvent volume is increased to 50 ml, a small peak appears around 10° in the PXRD pattern, which indicated the occurrence of MIL-118 in the products. The particle size distribution of crystal products was obtained under different solvent ratio conditions as shown in figure 6*d*, which shows that the particle size of the product increases with the amount of solvent. When the addition of solvent volume reached 40 ml, the particle size tended to decrease. Table 1 shows that the D50 is 15.25 µm, D90 is 24.79 µm and the average particle size is 15.17 µm.

The PXRD pattern of crystal products obtained under different addition amounts of sodium hydroxide is presented in figure 7*a*. The intensity and position of all the diffraction peaks of the three products with different sodium hydroxide additions were completely consistent with the as-synthesized patterns of the MIL-121, which proves that the three products have the same crystal form. The particle size distribution of crystal products prepared under the different addition amounts of sodium hydroxide shows that the particle size increases with the increase of the addition amounts of sodium hydroxide figure 7*d*. When the amount of sodium hydroxide solution exceeded 8 mmol, the reaction solution developed into gel-like consistency so that the particles could not separated by the centrifugation method.

Figure 8*a* shows the PXRD pattern of crystal products obtained under different addition amounts of 2-MI. The particle size distribution of crystal products obtained under different 2-MI addition amounts shows that the particle size of crystals increases with increase in the 2-MI addition amounts and then

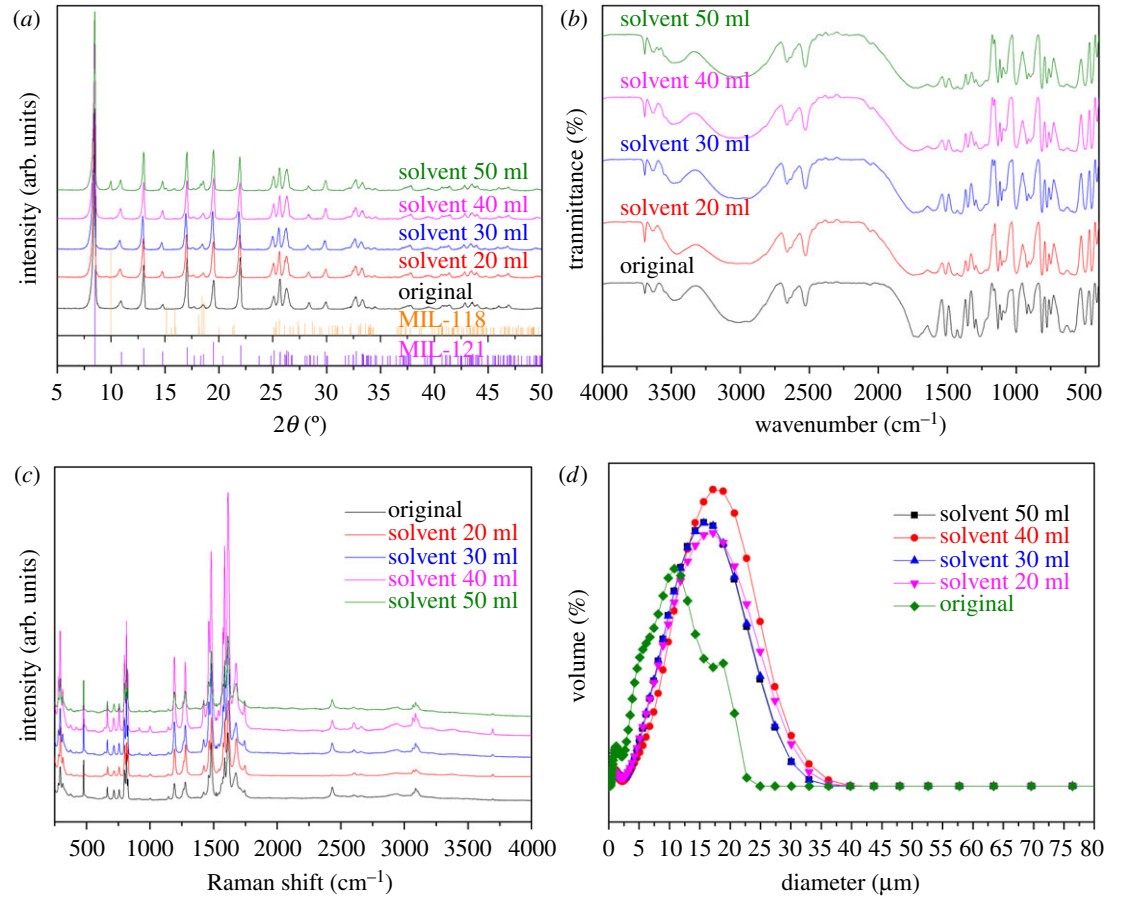

**Figure 6.** The PXRD, FT-IR, Raman and size distribution images of MIL-121 prepared at different amounts of solvent: (*a*) PXRD, (*b*) FT-IR, (*c*) Raman, and (*d*) size distribution.

**Table 1.** The particle size distribution of products under different addition amounts of solvent.

|  | original | 20 ml | 30 ml | 40 ml | 50 ml |
|---|---|---|---|---|---|
| mean (μm) | 8.56 | 14.17 | 13.09 | 15.17 | 13.13 |
| median (μm) | 7.96 | 13.79 | 13.08 | 15.25 | 13.14 |
| C.V. | 63.42 | 54.51 | 55.57 | 49.61 | 55.88 |
| D50 (μm) | 7.96 | 13.79 | 13.08 | 15.25 | 13.14 |
| D90 (μm) | 16.40 | 24.03 | 22.64 | 24.80 | 22.71 |

decreases figure 8*d*. When the addition amounts of 2-MI approach 8 mmol, a particle size distribution occurred after 32 μm, which is owing to crystal agglomeration after centrifugation and drying. The results coincide with the experimental phenomenon. When the amount of 2-MI is increased, the reaction solution is gel-like so that the powder cannot be separated by the centrifugation method.

## 3.3. Thermogravimetric analysis of five morphologies of MIL-121

The thermal stability of MIL-121 crystals with different morphologies was studied by thermogravimetric measurement. Thermogravimetric data analysis shows that the thermal behaviour of MIL-121 under $N_2$ atmosphere was divided into three stages [28]. Figure 9 shows that MIL-121 crystals with different morphologies have similar thermal behaviour. In the first step, the water molecules and pyromellitic acid in the channels were squeezed out. In the second step, the free carboxyl groups that were not involved in coordination between 350°C and 480°C were converted into carbon dioxide molecules. The third step was the decomposition of the framework between 480°C and 700°C. Comprehensive

**Figure 7.** The PXRD, FT-IR, Raman and size distribution images of MIL-121 prepared at different amounts of sodium hydroxide: (*a*) PXRD, (*b*) FT-IR, (*c*) Raman, and (*d*) size distribution.

analysis of the initial decomposition temperature, maximum weight loss temperature, maximum weight loss rate and residual mass percentage of samples with different morphologies can draw conclusions. The thermal stability of the five morphologies of MIL-121 crystals was good. The stacked sheet-shaped MIL-121 crystal obtained by changing the raw material ratio has the best stability, and the octahedral-shaped MIL-121 crystal obtained by adding 2-MI has the worst stability.

## 3.4. Analysis of MIL-121 crystallization mechanism

Hydrothermal reaction crystallization is a crystallization method that obtains supersaturation by coordination reaction. The crystallization centre is produced by the combination of metal ions and ligands under high-temperature and high-pressure conditions. Crystal growth was influenced by adjusting the hydrothermal reaction conditions. Typically, the crystallization process is divided into three periods [35]: (i) induction period, at this stage, particles of critical size were produced in the solution; (ii) nucleation period, during which critical particles evolve into crystal cores (primary nuclei); and (iii) growing period, solute molecules were arranged on the surface of the existing nucleus and then improved the growth of the crystals. This study system formed a stable supersaturated solution. As the temperature increased, the pressure in the reactor increased. The crystal core of the system is produced gradually by the combination of metal ions ($Al^{3+}$) and pyromellitic acid. As the supersaturation of the solution gradually decreases, the process of the transition of the complex to the crystal core slows down to the point where no new crystallization centre is produced. Zacher *et al.* [36] have reported that relatively slow and continuous nucleation processes and rapid particle growth have been observed during the nucleation of HKUST-1.

Also, the MOF crystal morphology is also known as crystal habit. It is determined by the crystal system, the unit cell parameters, the axial relationship of the crystal and the relative growth rate of the crystal. The symmetrical crystal structure is a basic parameter that determines its crystal habit. With the extension of synthesis time, the supersaturation and concentration of reactants decrease, which

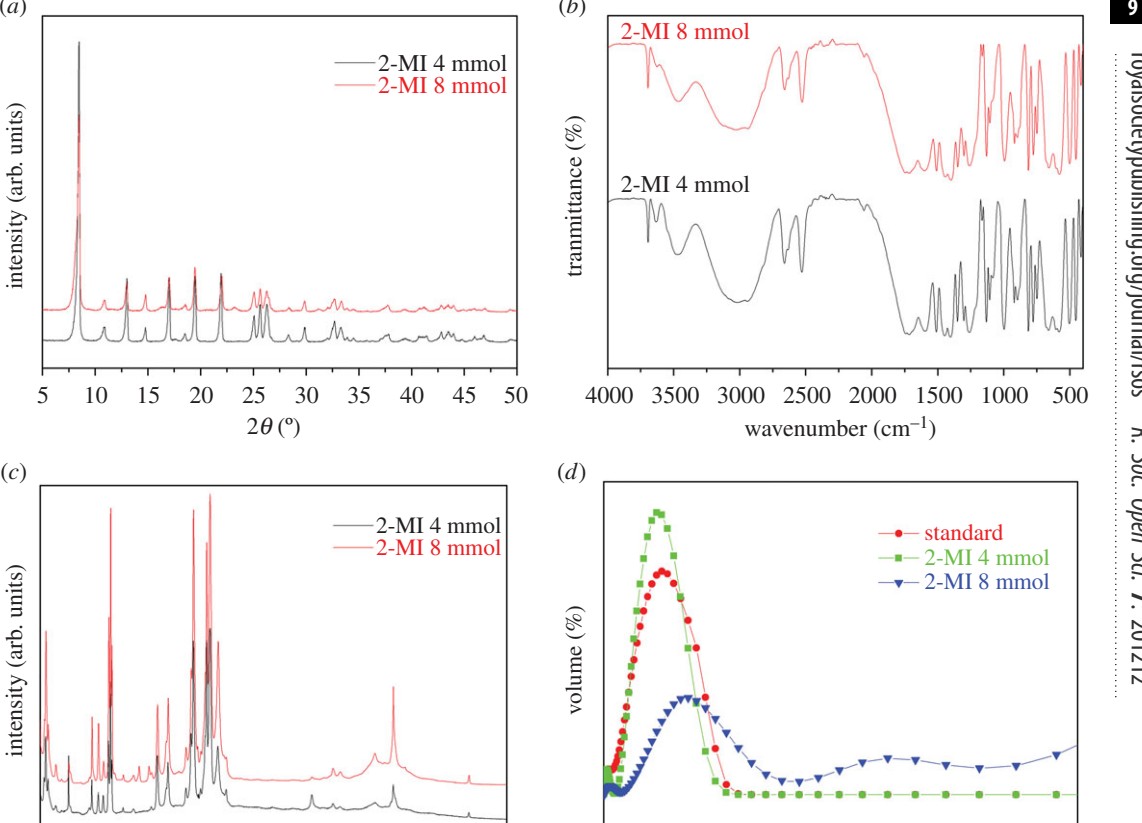

**Figure 8.** The PXRD, FT-IR, Raman and size distribution images of MIL-121 prepared at different amounts of 2-MI: (*a*) PXRD, (*b*) FT-IR, (*c*) Raman, and (*d*) size distribution.

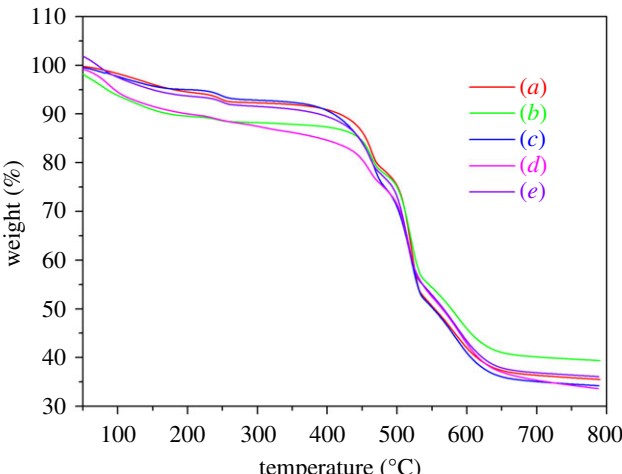

**Figure 9.** TGA curves of MIL-121 crystals of different morphologies: (*a*) solvent 40 ml, (*b*) NaOH 8 mmol, (*c*) LiCl 32 mmol, (*d*) 2-MI 8 mmol, and (*e*) $Al(NO_3)_3 \cdot 9H_2O : H_4BTEC : H_2O = 2 : 1 : 2000$.

result in the difference of the change in the growth rate of different crystal surfaces [35]. Generally speaking, there are three mechanisms of crystal growth [37]: diffusion-deposition growth mechanism, two-dimensional crystal core growth mechanism and surface defect growth mechanism. Small surfaces that grow quickly during the reaction are destined to disappear quickly, and those that grow slower will ultimately determine the final crystal habit [38].

It is worth emphasizing that the linear velocity of crystal surface growth increased with the supersaturation. Therefore, the crystals can grow rapidly under the conditions of hydrothermal reaction crystallization. Crystals growing rapidly may result in some irregular crystal and crystal agglomeration. The increased solvent volume will lead to a decrease in the supersaturation of the solution. Lower nucleation rate is prone to produce larger particles of MIL-121 crystals. The exposure of different crystal planes of MIL-121 is controlled by a simple hydrothermal route plus sodium hydroxide to adjust the pH of the mixed solution. The addition amount of sodium hydroxide causes the exposure of different crystal planes and the growth rate changes to produce the final irregular morphology. The facet control was realized by adjusting the pH of the solution by the addition of NaOH [39]. Lithium salt as an additive also produces the final morphology by adjusting the growth rate of different crystal planes. 2-MI was an important addictive during the hydrothermal process ascribing to its nitrogen heterocycle group, which can strongly chelate metal ions. Consequently, the decline of free metal ions would slow down the nucleation rate and crystal growth of precursors during the hydrothermal process, thus affecting the morphologies of the ultimate products. Control experiments indicated that the crystal nucleation and growth were affected by both coordination and deprotonation equilibria of 2-MI [40]. As observed in the experimental results, the addition of sodium hydroxide tends to obtain irregularly blocky crystals, the occurrence of lithium chloride will generate crystals with a column shape, the presence of 2-MI makes the obtained crystals octahedral-like and the adjustment of raw ratio has a tendency to develop a stacked sheet crystal.

# 4. Conclusion

In this work, the effects of hydrothermal reaction conditions on MIL-121 synthesis were studied. The morphology of MIL-121 is controlled by using sodium hydroxide, lithium chloride and 2-MI as additives. The resulting solids were analysed with SEM, PXRD, FT-IR, Raman and a laser diffraction-based particle size analyser. Five morphologies of MIL-121 crystal were observed by SEM. The MIL-121 crystal size differences in five morphologies were analysed by an LS13320 laser particle size analyser. When the amount of solvent was 40 ml, the crystal habit of MIL-121 was an irregular cube. Sodium hydroxide was added to adjust the pH of the reaction solution to 1.67 and the crystal habit of MIL-121 was irregular and blocky. The optimal amount of lithium chloride is 32 mmol, and the crystal habit of MIL-121 was columnar. Competitive ligand 2-MI was added in an amount of 8 mmol, and the crystal appeared as an octahedral morphology. By adjusting the ratio of raw materials, the crystal appeared as a stacked sheet morphology. The use of lithium salts and sodium hydroxide as additives will provide alternative strategies for the adjustment of other MOF morphologies. The MIL-121 crystals with different morphologies have different sizes and thermal stability. This provides a choice for it as a new generation of MOF adsorbent, which is expected to achieve selective adsorption of lithium ions.

Data accessibility. The data support related to this article has been uploaded as part of the electronic supplementary material.

Authors' contributions. F.W. was involved in the formal analysis of date and drafting the article; L.Z. was involved in revising it critically for important intellectual content; Q.W. was involved in investigation and acquisition of date; Y.W. was involved in visualization and supervision.

Competing interests. We declare we have no competing interests.

Funding. We received no funding for this study.

Acknowledgements. The authors are thankful to the Industrial Crystallization and Particle Laboratory of Tianjin University of Science and Technology for providing the experimental platform and financial support of this study.

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
