## [Reviewer comments · Royal Society Open Science]

Review History

RSOS-201212.R0 (Original submission)

Review form: Reviewer 1

Is the manuscript scientifically sound in its present form?

Yes

Are the interpretations and conclusions justified by the results?

Yes

Is the language acceptable?

No

Do you have any ethical concerns with this paper?

No

Have you any concerns about statistical analyses in this paper?

No

Recommendation?

Accept with minor revision (please list in comments)

Comments to the Author(s)

The manuscript "Research on the Effects of Hydrothermal Synthesis Conditions on the Crystal Habit of MIL-121" by Zhu and coworkers deals with the formation of MIL MOFs and the modulation of the crystal morphology by adjusting the synthesis process. It shows interesting results and should be published after revision. In my opinion the manuscript should be spell checked thoroughly.

General comments:

1. On page 5: The authors state that the crystal habit does not change with temperature (last sentence of the first paragraph), but two sentences before it is written that the crystal phase changes.
2. In my opinion Figure 4 could be improved by showing the comparison of the different concentrations and conditions (to be moved from the SI) and moved the lower magnification images to the SI. The comparison is more important than the magnification to understand the results.
3. Some literature from the field of MOF crystal modulation that could be interesting:
<https://onlinelibrary.wiley.com/doi/full/10.1002/ange.202000795>
<https://pubs.acs.org/doi/abs/10.1021/jacs.7b12633>
<https://www.nature.com/articles/s41467-020-14671-9>
<https://pubs.rsc.org/en/content/articlehtml/2020/cs/c9cs00871c>
<https://chemistry-europe.onlinelibrary.wiley.com/doi/full/10.1002/slct.201803971>
<https://www.publish.csiro.au/CH/CH19271>
4. I am a bit puzzled about Figure 5 that shows one example for all conditions. Only minor changes are observed and the results are shown in the next Figures anyway. Maybe cut this Figure or just show the size difference?
5. In my opinion, the data in Table 1 seems to be too accurate
6. I would suggest to mention some possible applications in the conclusion.
7. One of the major points that need revision is the discussion about the reasons for the observed behavior. At the moment there are only small hints in the manuscript. As the authors used a variety of additives, a thorough explanation should be added for the different additives.

Specific comments:

8. The first sentence of the Summary does not really make sense
9. References 15-17 are not superscript
10. Page 5: "was obtained under different", I guess PXRD was not obtained at different temperatures, but the synthesis was performed at different temperatures
11. I suggest to add a dotted line for the major peaks in PXRD to make comparison with literature easier (especially in Figure 3)
12. The scale bars for SEM are hard to read, maybe add new one that are larger? Also the text in the SEM images (e.g. a) 120 °C) is hard to read, maybe change to white bold text?
13. The y-axis description in Figure 2 is missing
14. Page 9: typo "tFhat"
15. Page 11: "nuclearzation" maybe it should be "nucleation"?

16. The SI should also have the paper title and authors.
17. The page numbers in the SI should be "SX"
18. The order of Figures in the SI is strange (9 before 2) and the Figures missing (Fig S1, and S3-S9)

Review form: Reviewer 2

Is the manuscript scientifically sound in its present form?

Yes

Are the interpretations and conclusions justified by the results?

Yes

Is the language acceptable?

No

Do you have any ethical concerns with this paper?

No

Have you any concerns about statistical analyses in this paper?

No

Recommendation?

Accept with minor revision (please list in comments)

Comments to the Author(s)

In the manuscript the authors report investigations on the influence of hydrothermal synthesis conditions on morphology and size distribution of MIL-121 crystals. Synthesis parameters like temperature, amount of solvent, type of additives like sodium hydroxide, lithium chloride or 2-methylimidazole were varied. The products were characterized with SEM, XRD, FTIR and determination of particle size distribution. Such investigations are useful for understanding reaction parameters on crystal growth, size and size distribution. Hence I'll recommend acceptance of the paper after some corrections/additions.

The authors should make a comment why these additives were selected. The instrument used for recording Raman spectra should be added.

Why is the first Figure in Supporting Information Figure S9 and not Figure S1?

On p.5 the authors mention Figure 2a but there is no such figure. The authors should also shortly discuss what is seen in Figure 2. It seems that at the smallest particles are obtained at 120 °C and that T = 210 °C afforded a bimodal size distribution. This should be discussed.

On several places the other mention that the crystals were caked. What does it mean?

Agglomerated?

Decision letter (RSOS-201212.R0)

Dear Mr Wang:

Title: Research on the Effects of Hydrothermal Synthesis Conditions on the Crystal Habit of MIL-121

Manuscript ID: RSOS-201212

Thank you for submitting the above manuscript to Royal Society Open Science. On behalf of the Editors and the Royal Society of Chemistry, I am pleased to inform you that your manuscript will be accepted for publication in Royal Society Open Science subject to minor revision in accordance with the referee suggestions. Please find the reviewers' comments at the end of this email.

The reviewers and handling editors have recommended publication, but also suggest some minor revisions to your manuscript. Therefore, I invite you to respond to the comments and revise your manuscript.

Because the schedule for publication is very tight, it is a condition of publication that you submit the revised version of your manuscript before 01-Oct-2020. Please note that the revision deadline will expire at 00.00am on this date. If you do not think you will be able to meet this date please let me know immediately.

Kind regards,
Dr Laura Smith
Publishing Editor, Journals

On behalf of the Subject Editor Professor Anthony Stace and the Associate Editor Dr Annette Trunschke.

RSC Associate Editor:
Comments to the Author:
(There are no comments.)

RSC Subject Editor:
Comments to the Author:
(There are no comments.)

Reviewer comments to Author:
Reviewer: 1

Comments to the Author(s)

The manuscript "Research on the Effects of Hydrothermal Synthesis Conditions on the Crystal Habit of MIL-121" by Zhu and coworkers deals with the formation of MIL MOFs and the modulation of the crystal morphology by adjusting the synthesis process. It shows interesting results and should be published after revision. In my opinion the manuscript should be spell checked thoroughly.

General comments:

1. On page 5: The authors state that the crystal habit does not change with temperature (last sentence of the first paragraph), but two sentences before it is written that the crystal phase changes.
2. In my opinion Figure 4 could be improved by showing the comparison of the different concentrations and conditions (to be moved from the SI) and moved the lower magnification images to the SI. The comparison is more important than the magnification to understand the results.
3. Some literature from the field of MOF crystal modulation that could be interesting:
<https://onlinelibrary.wiley.com/doi/full/10.1002/ange.202000795>
<https://pubs.acs.org/doi/abs/10.1021/jacs.7b12633>
<https://www.nature.com/articles/s41467-020-14671-9>

<https://pubs.rsc.org/en/content/articlehtml/2020/cs/c9cs00871c>

<https://chemistry-europe.onlinelibrary.wiley.com/doi/full/10.1002/slct.201803971>

<https://www.publish.csiro.au/CH/CH19271>

4. I am a bit puzzled about Figure 5 that shows one example for all conditions. Only minor changes are observed and the results are shown in the next Figures anyway. Maybe cut this Figure or just show the size difference?

5. In my opinion, the data in Table 1 seems to be too accurate

6. I would suggest to mention some possible applications in the conclusion.

7. One of the major points that need revision is the discussion about the reasons for the observed behavior. At the moment there are only small hints in the manuscript. As the authors used a variety of additives, a thorough explanation should be added for the different additives.

Specific comments:

8. The first sentence of the Summary does not really make sense

9. References 15-17 are not superscript

10. Page 5: "was obtained under different", I guess PXRD was not obtained at different temperatures, but the synthesis was performed at different temperatures

11. I suggest to add a dotted line for the major peaks in PXRD to make comparison with literature easier (especially in Figure 3)

12. The scale bars for SEM are hard to read, maybe add new one that are larger? Also the text in the SEM images (e.g. a) 120 °C) is hard to read, maybe change to white bold text?

13. The y-axis description in Figure 2 is missing

14. Page 9: typo "tFhat"

15. Page 11: "nuclearzation" maybe it should be "nucleation"?

16. The SI should also have the paper title and authors.

17. The page numbers in the SI should be "SX"

18. The order of Figures in the SI is strange (9 before 2) and the Figures missing (Fig S1, and S3-S9)

Reviewer: 2

Comments to the Author(s)

In the manuscript the authors report investigations on the influence of hydrothermal synthesis conditions on morphology and size distribution of MIL-121 crystals. Synthesis parameters like temperature, amount of solvent, type of additives like sodium hydroxide, lithium chloride or 2-methylimidazole were varied. The products were characterized with SEM, XRD, FTIR and determination of particle size distribution. Such investigations are useful for understanding reaction parameters on crystal growth, size and size distribution. Hence I'll recommend acceptance of the paper after some corrections/additions.

The authors should make a comment why these additives were selected. The instrument used for recording Raman spectra should be added.

Why is the first Figure in Supporting Information Figure S9 and not Figure S1?

On p.5 the authors mention Figure 2a but there is no such figure. The authors should also shortly discuss what is seen in Figure 2. It seems that at the smallest particles are obtained at 120 °C and that T = 210 °C afforded a bimodal size distribution. This should be discussed.
On several places the other mention that the crystals were caked. What does it mean?
Agglomerated?

Author's Response to Decision Letter for (RSOS-201212.R0)

See Appendix A.

Decision letter (RSOS-201212.R1)

Dear Mr Wang:

Title: Research on the Effects of Hydrothermal Synthesis Conditions on the Crystal Habit of MIL-121

Manuscript ID: RSOS-201212.R1

It is a pleasure to accept your manuscript in its current form for publication in Royal Society Open Science. The chemistry content of Royal Society Open Science is published in collaboration with the Royal Society of Chemistry.

On behalf of the Subject Editor Professor Anthony Stace and the Associate Editor Dr Annette Trunschke.

RSC Associate Editor

Comments to the Author:

The corresponding author replied to all queries of the Reviewers and improved the manuscript accordingly. I recommend to accept.

Reviewer(s)' Comments to Author:

Appendix A

List of Responses

[Date] 2020.09.27

Dear Editors and Reviewers:

Thank you for your letter and for the reviewers' comments concerning our manuscript entitled **"Research on the Effects of Hydrothermal Synthesis Conditions on the Crystal Habit of MIL-121"** (ID: RSOS-201212). Those comments are all valuable and very helpful for revising and improving our paper, as well as the important guiding significance to our researches. We have studied comments carefully and have made correction which we hope meet with approval. Revised portion are marked in blue in the paper. The main corrections in the paper and the responds to the reviewer's comments are as flowing:
The font color and paragraph format have been revised in the article.

The content of the article has been mainly revised in five parts:

First, the characterization method test part, Raman and thermogravimetric test methods and test conditions are added. Second, the analysis description in the first paragraph of the results and discussion has been reorganized and revised, and the analysis of Figure 2 has been added. Third, the SEM image in the article has been modified and rearranged to make it clearer, highlighting the process of additive concentration and reaction conditions affecting morphology changes. Fourth, the thermal behavior analysis of different morphologies of MIL-121 crystals has been added to the structure and discussion section, and related potential applications are pointed out in the conclusion section. Fifth, the content and pictures of the supporting information have been modified, and the cover has been added, including the name, title, author and other information of the journal. Finally, the content of the article was carefully checked for spelling and formatting.

Responds to the reviewer's comments:

Reviewer: 1

Comments to the Author(s)

The manuscript "Research on the Effects of Hydrothermal Synthesis Conditions on the Crystal Habit of MIL-121" by Zhu and coworkers deals with the formation of MIL MOFs and the modulation of the crystal morphology by adjusting the synthesis process. It shows interesting results and should published after revision. In my opinion the manuscript should be spell checked thoroughly.

Response: Thank you very much for your affirmation of the research work of this article. We have carefully checked the spelling of the manuscript. The revised content is traced and annotated in the manuscript in blue.

General comments:

1. On page 5: The authors state that the crystal habit does not change with temperature (last sentence of the first paragraph), but two sentences before it is written that the crystal phase changes.

Response: Special thanks to you for your good comment. Yes. It can be seen from Figure 3 that the synthesis temperature is below 180 °C, and the PXRD pattern of the product is consistent with the theoretical pattern of MIL-121, and the morphology does not change much. Although the morphology changes after the temperature is higher than 180 °C, a new crystal form of MIL-118 appears. It can be seen from the PXRD comparison that it is a mixture of the two, which is not the main research content of this article. I have revised the description of this paragraph and added an analysis of the particle size distribution in Figure 2.

2. In my opinion Figure 4 could be improved by showing the comparison of the different concentrations and conditions (to be moved from the SI) and moved the lower magnification images to the SI. The comparison is more important than the magnification to understand the results.

Response: Special thanks to you for your good comment. I have reorganized the original Figure 4, and added new Figures 4 and 5 to reflect the changes in the topography. Mainly compare the influence of different concentration and reaction conditions on the morphology. Put the original low-magnification image into the SI.

3. Some literature from the field of MOF crystal modulation that could be interesting:

<https://onlinelibrary.wiley.com/doi/full/10.1002/ange.202000795>

<https://pubs.acs.org/doi/abs/10.1021/jacs.7b12633>

<https://www.nature.com/articles/s41467-020-14671-9>
<https://pubs.rsc.org/en/content/articlehtml/2020/cs/c9cs00871c>
<https://chemistry-europe.onlinelibrary.wiley.com/doi/full/10.1002/slct.201803971>
<https://www.publish.csiro.au/CH/CH19271>

Response: Yes, your suggestion is very good. The articles on MOF morphology control you recommend are very interesting. The use of dual-hydrophilic block copolymers, polyacrylic acid and surfactants as modulators to control the morphology of MOF has many advantages and is some emerging control methods. This will provide ideas and references for me to continue to study MOF morphology and size control. I will be inspired from this and expand the practical application of different shape control.

4. I am a bit puzzled about Figure 5 that shows one example for all conditions. Only minor changes are observed and the results are shown in the next Figures anyway. Maybe cut this Figure or just show the size difference?

Response: Special thanks to you for your good comment. According to my previous ideas, five different morphologies were screened through different additive adjustments, and the characterization data of the product obtained under the optimal additive amount condition was drawn in Figure 5. Indeed, it is more important to investigate process changes. Thank you very much for your suggestion. I have modified this part of the content. Figure 5 and related descriptions have been moved to SI. SEM images with different concentrations and conditions affecting the topography have been added to the manuscript and a new Figure 5 has been drawn.

5. In my opinion, the data in Table 1 seems to be too accurate

Response: Special thanks to you for your good comment. The data in Figure 1 was measured with a Laser diffraction particle size analyzer (BECKMAN COULTER LS). It does seem to be too accurate. I have corrected it in the manuscript to keep 2 significant digits after the decimal point. At the same time, Table S2, Table S3, Table S4, Table S5, and Table S6 are used in SI to list the raw data of product particle size measurement under different solvent content conditions.

6. I would suggest to mention some possible applications in the conclusion.

Response: Special thanks to you for your good comment. At present, we are actively studying some possible applications of MIL-121 with different morphologies, and try to load other substances in MOF channels for ion adsorption research. We have added possible applications in the conclusion section according to your suggestions.

7. One of the major points that need revision is the discussion about the reasons for the observed behavior. At the moment there are only small hints in the manuscript. As the authors used a variety of additives, a thorough explanation should be added for the different additives.

Response: Special thanks to you for your good comment. There are still many shortcomings in our research on how additives affect the morphology of MOF in the article. At present, we are trying to study the growth mechanism of MIL-121 with different morphologies through AFM and other means, but no great progress has been made. We have quoted some articles in the article to explain the reasons for this phenomenon, and hope to help you. Salting is only a result of different crystal surface growth rate changes to produce different crystal habits. There are many articles on the use of salt as an additive to regulate the morphology. For example, Qin et al. reported that sodium salt was utilized as additive to manipulate morphology and complex permittivity of NiCo_2O_4 absorbers through hydrothermal method (Applied Surface Science, 2020, 504, 144480). About adding 2-MI, Guo et al. in the article described that "Control experiments indicated that the crystal nucleation and growth were affected by both coordination and deprotonation equilibria of 2-MI. The 2-MI not only acts as competitive ligands at the metal centers but also serves as the base for the deprotonation of the bridging ligands. Thus, it will affect crystal nucleation and growth via both coordination and deprotonation equilibria." (Royal Society of Chemistry, 2018, 54, 252-255)

Specific comments:

8. The first sentence of the Summary does not really make sense

Response: Thank you very much for your suggestion. We have corrected the manuscript and deleted the first sentence in the abstract.

9. References 15-17 are not superscript

Response: Thank you very much for your suggestion. We have made corrections in the manuscript and changed references 15-17 to superscript.

10. Page 5: "was obtained under different", I guess PXRD was not obtained at different temperatures, but the synthesis was performed at different temperatures

Response: Yes, as you might guess. What we want to express is the PXRD patterns of crystal products obtained under different synthesis temperature conditions. We are very sorry that our incorrect description has caused you trouble. We have corrected this description in the manuscript.

11. I suggest to add a dotted line for the major peaks in PXRD to make comparison with literature easier (especially in Figure 3)

Response: Special thanks to you for your good comment. We have added a dotted line to the PXRD in Figure 3(a) according to your suggestion for comparison with the theoretical PXRD. Other PXRD patterns are just different concentrations and conditions for comparison without adding dotted lines.

12. The scale bars for SEM are hard to read, maybe add new one that are larger? Also the text in the SEM images (e.g. a) 120 °C) is hard to read, maybe change to white bold text?

Response: Special thanks to you for your good comment. We have drawn the scale of the new SEM image in the manuscript and changed the text color to white according to your suggestion. For the SEM in SI, the original scale is retained, and the text color is changed to white.

13. The y-axis description in Figure 2 is missing

Response: We have redrawn Figure 2 and added the description of the y-axis. The replacement in the manuscript has been completed.

14. Page 9: typo "tFhat"

Response: Yes, "that" is a typing error. We are very sorry for our negligence of wrong typo.

15. Page 11: "nuclearzation" maybe it should be "nucleation"?

Response: Yes, " nuclearzation " is a spelling error. We are very sorry for our negligence of wrong spell.

16. The SI should also have the paper title and authors.

Response: Special thanks to you for your good comment. We have revised the content and format of SI and added some new test data. Added journal name, title, author and other information on the first page.

17. The page numbers in the SI should be "SX"

Response: Special thanks to you for your good comment. We have rearranged the SI page numbers according to your suggestions.

18. The order of Figures in the SI is strange (9 before 2) and the Figures missing (Fig S1, and S3-S9)

Response: In the original SI, the row starting from S9 is next to Figure 8 in the manuscript. The appearance of S2 is due to our negligence. This sorting method is indeed incorrect. We have revised the SI content and reordered the graphs and tables.

Reviewer: 2

Comments to the Author(s)

In the manuscript the authors report investigations on the influence of hydrothermal synthesis conditions on morphology and size distribution of MIL-121 crystals. Synthesis parameters like temperature, amount of solvent, type of additives like sodium hydroxide, lithium chloride or 2- methylimidazole were varied. The products were characterized with SEM, XRD, FTIR and determination of particle size distribution. Such investigations are useful for understanding reaction parameters on crystal growth, size and size distribution. Hence I'll recommend acceptance of the paper after some corrections/additions.

Response: Thank you very much for your affirmation of the research work of this article. We have carefully checked the spelling of the manuscript. The revised content is traced and annotated in the manuscript in blue.

1.The authors should make a comment why these additives were selected.

Response: Special thanks to you for your good comment. Before conducting the experiment, I read a lot of literature on MOF morphology control, and obtained the following control methods through the literature summary: 1. Change the solvent; 2. Adjust the reaction temperature; 3. Add template or regulator; 4. Add 2- Methyl imidazole. There are also some emerging control methods like microwave assist, laser induction, gravity method, ionic liquid and so on. Combining the MIL-121 system I selected, and on this basis, I decided to change the amount of solvent, add 2-MI and other green control methods. At the same time, it was found that no common metal salt was used to adjust the morphology of MOF, so sodium hydroxide without impurities in the system was selected as an additive. In addition, our ultimate goal is to achieve selective adsorption of lithium ions, so we chose lithium chloride as an additive to modify MIL-121, but unexpectedly found that its morphology changed. Subsequent surface scans revealed no lithium chloride residues in the final product.

2.The instrument used for recording Raman spectra should be added.

Response: Special thanks to you for your good comment. The Raman detection was carried out on a LabRAM HR800 microprobe Raman system (Horiba Jobin-Yvon) with excitation of 532 nm. We have added Raman spectroscopy recording equipment to the manuscript. At the same time, the thermogravimetric test data and thermal stability analysis are supplemented in the article.

3.Why is the first Figure in Supporting Information Figure S9 and not Figure S1?

Response: In the original SI, the row starting from S9 is next to Figure 8 in the manuscript. This sorting method is indeed incorrect. We have revised the SI content and reordered the graphs and tables.

4.On p.5 the authors mention Figure 2a but there is no such figure.

Response: Yes, it should be Figure 3a. We are very sorry for our negligence of wrong number ,we have made correction according to the Reviewer's comments.

5.The authors should also shortly discuss what is seen in Figure 2. It seems that at the smallest particles are obtained at 120 °C and that T = 210 °C afforded a bimodal size distribution. This should be discussed.

Response: Special thanks to you for your good comment. We have added a discussion of Figure 2 to the manuscript. At 210 °C, the crystal size distribution appears in a dual-mode model distribution. This is because when the temperature is higher than 180 °C, the crystal form of MIL-121 is transformed into MIL-118, and the product exists in the form of a mixture of the two.

6.On several places the other mention that the crystals were caked. What does it mean? Agglomerated?

Response: Yes, it is agglomeration. We have made corrections in the manuscript. What we want to express is that the crystals cannot be separated to obtain products for related characterization due to agglomeration. It is hard to be ground after drying.

Special thanks to you for your good comments.

We tried our best to improve the manuscript and made some changes in the manuscript. And here we did not list the changes but marked in blue in revised paper.

We appreciate for Editors/Reviewers' warm work earnestly, and hope that the correction will meet with approval.

Once again, thank you very much for your comments and suggestions.

Sincerely,

[Corresponding authors' name] Liang Zhu

[Affiliation] College of Chemical Engineering and Materials Science, Tianjin University of Science & Technology